# Affect-focused psychodynamic psychotherapy for depression and anxiety through the Internet: a randomized controlled trial

Robert Johansson[1], Martin Björklund[1], Christoffer Hornborg[1], Stina Karlsson[1], Hugo Hesser[1], Brjánn Ljótsson[2], Andréas Rousseau[3], Ronald J. Frederick[4] and Gerhard Andersson[1,5,6]

[1] Department of Behavioural Sciences and Learning, Linköping University, Linköping, Sweden
[2] Department of Clinical Neuroscience, Division of Psychology, Karolinska Institutet, Sweden
[3] Psychiatric Clinic, University Hospital of Linköping, Linköping, Sweden
[4] Center for Courageous Living, California, United States
[5] Department of Clinical Neuroscience, Psychiatry Section, Karolinska Institutet, Stockholm, Sweden
[6] Swedish Institute for Disability Research, Linköping University, Linköping, Sweden

Corresponding author
Robert Johansson,
robert.johansson@liu.se

## ABSTRACT

**Background.** Psychodynamic psychotherapy is a psychological treatment approach that has a growing empirical base. Research has indicated an association between therapist-facilitated affective experience and outcome in psychodynamic therapy. Affect-phobia therapy (APT), as outlined by McCullough et al., is a psychodynamic treatment that emphasizes a strong focus on expression and experience of affect. This model has neither been evaluated for depression nor anxiety disorders in a randomized controlled trial. While Internet-delivered psychodynamic treatments for depression and generalized anxiety disorder exist, they have not been based on APT. The aim of this randomized controlled trial was to investigate the efficacy of an Internet-based, psychodynamic, guided self-help treatment based on APT for depression and anxiety disorders.

**Methods.** One hundred participants with diagnoses of mood and anxiety disorders participated in a randomized (1:1 ratio) controlled trial of an active group versus a control condition. The treatment group received a 10-week, psychodynamic, guided self-help treatment based on APT that was delivered through the Internet. The treatment consisted of eight text-based treatment modules and included therapist contact (9.5 min per client and week, on average) in a secure online environment. Participants in the control group also received online therapist support and clinical monitoring of symptoms, but received no treatment modules. Outcome measures were the 9-item Patient Health Questionnaire Depression Scale (PHQ-9) and the 7-item Generalized Anxiety Disorder Scale (GAD-7). Process measures were also included. All measures were administered weekly during the treatment period and at a 7-month follow-up.

**Results.** Mixed models analyses using the full intention-to-treat sample revealed significant interaction effects of group and time on all outcome measures, when comparing treatment to the control group. A large between-group effect size of Cohen's $d = 0.77$ (95% CI: 0.37–1.18) was found on the PHQ-9 and a moderately

large between-group effect size $d = 0.48$ (95% CI: 0.08–0.87) was found on the GAD-7. The number of patients who recovered (had no diagnoses of depression and anxiety, and had less than 10 on both the PHQ-9 and the GAD-7) were at post-treatment 52% in the treatment group and 24% in the control group. This difference was significant, $\chi^2(N = 100, df = 1) = 8.3, p < .01$. From post-treatment to follow-up, treatment gains were maintained on the PHQ-9, and significant improvements were seen on the GAD-7.

**Conclusion.** This study provides initial support for the efficacy of Internet-delivered psychodynamic therapy based on the affect-phobia model in the treatment of depression and anxiety disorders. The results support the conclusion that psychodynamic treatment approaches may be transferred to the guided self-help format and delivered via the Internet.

## INTRODUCTION

The aim of this randomized controlled trial was to investigate the efficacy of an Internet-delivered psychodynamic guided self-help treatment for depression and anxiety disorders that was based on the affect-phobia model of psychopathology (*McCullough et al., 2003*). The project extends previous research on Internet-delivered psychological treatments in general, and that of Internet-delivered psychodynamic psychotherapy in particular (*Andersson et al., 2012*; *Johansson et al., 2012*). An overview of the trial can be seen in Fig. 1.

Depression and anxiety disorders are major world-wide health problems which lower the quality of life for the individual and generate large costs for society (*Ebmeier, Donaghey & Steele, 2006*; *Smit et al., 2006*). Lifetime prevalence for mood disorders and anxiety disorders in the US have been estimated to be 20.8% and 28.8%, respectively (*Kessler et al., 2005*).

Psychodynamic psychotherapy is a psychological treatment approach that has a growing empirical base (*Town et al., 2012*), with research support for e.g., depression (*Driessen et al., 2010*), social anxiety disorder (*Leichsenring et al., 2013*), panic disorder (*Milrod et al., 2007*), and generalized anxiety disorder (*Leichsenring et al., 2009*). There is a variation among the psychodynamic therapies in the degree to which they focus on expression and experience of affect. *Diener, Hilsenroth & Weinberger (2007)* conducted a meta-analysis of high-quality studies that had examined the role of therapist focus on affect in psychodynamic psychotherapy. The results indicated that the more therapists facilitated the affective experience/expression in psychodynamic therapy, the more patients improved (*Diener, Hilsenroth & Weinberger, 2007*). Thus, keeping a focus on affect may be one way of enhancing psychodynamic psychotherapies.

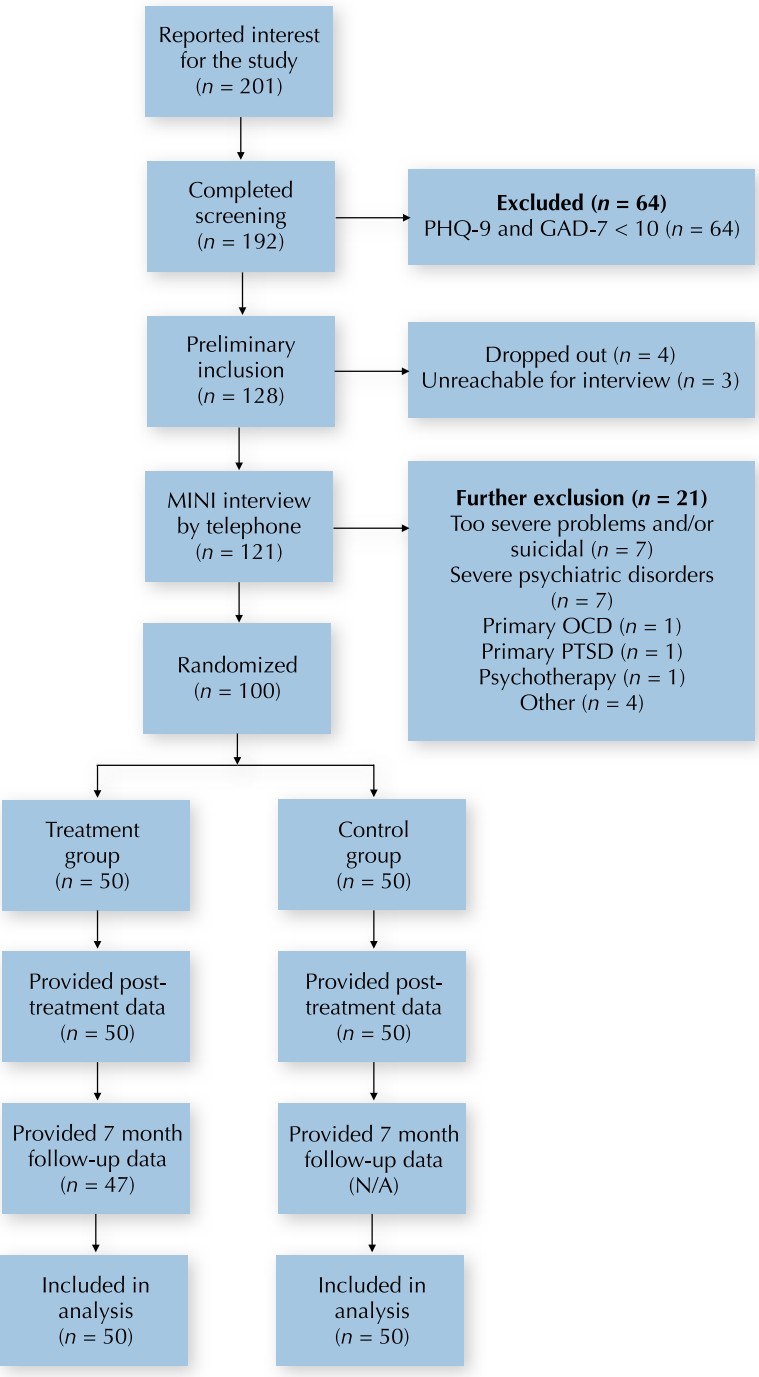

**Figure 1  CONSORT flowchart.**

One psychodynamic treatment that has a strong focus on expression and experience of affect is affect-phobia therapy (APT), developed by *McCullough et al. (2003)*. APT follows a treatment model which adheres to the fundamental structure of psychodynamic psychotherapy as outlined by Malan's triangle of conflict (i.e., the experience/expression

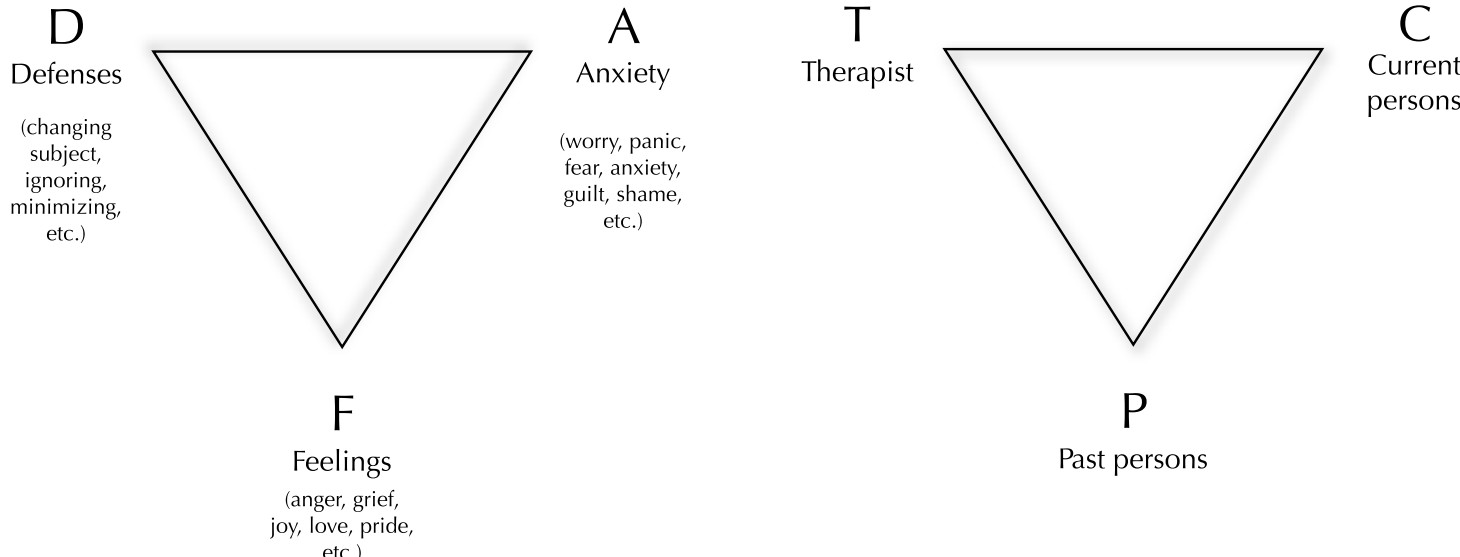

**Figure 2 Malan's two triangles - the triangle of conflict and the triangle of person.** The two triangles (*Malan, 1995*) represent what David Malan called "the universal principle of psychodynamic psychotherapy". That is, defenses (D) and anxieties (A) can block the expression of true feelings (F). These patterns began with past persons (P), are maintained with current persons (C), and are often enacted with the therapist (T).

of feelings (F) is blocked by defenses (D) and anxieties (A)) and triangle of person (i.e., conflicted patterns began with past persons (P), are maintained with current persons (C), and can be enacted with a therapist (T)), as illustrated in Fig. 2 (*Malan, 1995*). Typically in APT, the therapist clarifies a client's defenses, helps the client to observe and experience the underlying affects, and helps the client to regulate associated anxiety (*McCullough et al., 2003*). Formally, the treatment includes three main treatment objectives: defense restructuring (recognizing and relinquishing maladaptive defenses), affect restructuring (desensitization of affects through exposure to conflicted feeling), and self/other restructuring (improvement in sense of self and relationship with others). The main goal of psychodynamic psychotherapy based on the APT model is to help clients experience and to adaptively express previously avoided feelings (*McCullough et al., 2003*). That goal is shared with an entire set of psychodynamic psychotherapies that are grouped under the umbrella term *experiential dynamic therapies* (*Osimo & Stein, 2012*), which in addition to APT includes, for example, Intensive Short-Term Dynamic Psychotherapy (*Abbass, Town & Driessen, 2012*; *Davanloo, 2000*), and Accelerated Experiential Dynamic Psychotherapy (*Fosha, 2000*). Two randomized trials, investigating the efficacy of APT in the treatment of personality disorders, found that APT can be effective in reducing general psychiatric symptoms (*Svartberg, Stiles & Seltzer, 2004*; *Winston et al., 1994*). However, except for case-series and some small uncontrolled studies (e.g., *Dornelas et al., 2010*), to date no trial has investigated the efficacy of APT for patients with a principal Axis I disorder.

During the last decade, numerous trials on guided self-help and Internet-delivered cognitive behavior therapy (CBT) for various psychiatric disorders have been conducted (*Andersson, 2009*; *Hedman, Ljótsson & Lindefors, 2012*; *Johansson & Andersson, 2012*). For

mild to moderate depression and anxiety disorders, it seems safe to conclude that these treatments are as effective as face-to-face treatments (*Cuijpers et al., 2010*). While most research regarding Internet-based psychological treatments have concerned CBT, there are exceptions. Results from two recent randomized controlled trials focusing on the treatment of depression and generalized anxiety disorder indicate that psychodynamic treatments can also be delivered via the Internet (*Andersson et al., 2012*; *Johansson et al., 2012*).

This randomized controlled trial aimed to examine the effects of an Internet-delivered psychodynamic treatment based on the affect-phobia model of psychopathology. Participants had depression and anxiety disorders. The treatment was given as self-help with additional therapist support via the Internet, and compared to a control group who also received online support. As compared with the control condition, a significant effect of treatment was expected both on measures of depression and anxiety for the full sample. In addition, a larger effect was expected on measures of depression for participants with depression as their main presenting problem as compared with those who did not have this as the main problem. Similar, a larger effect on anxiety measures was expected for participants with a principal anxiety diagnosis as compared with those who did not have such a diagnosis. We also investigated the uncontrolled effects of the treatment 7 months following the completion of the treatment.

## MATERIALS & METHODS

This study is reported in accordance with the CONSORT statement for clinical trials (*Schulz, Altman & Moher, 2010*). The clinicaltrials.gov registration ID is NCT01532219. This study received approval from the Regional Ethics Board of Linköping, Sweden (Approval number: 2011/400-31). Written informed consent was obtained from all participants via the online treatment platform. Participants received the treatment at no cost. After being enrolled in the study, all participants were assigned one of the therapists as their personal contact. Half of the participants received psychodynamic treatment in the format of guided self-help and the other half was assigned to a waiting-list where participants also received support via the Internet. The waiting-list served as the control group.

### Participants

Patients were recruited via the Internet and advertisements in newspapers during January 2012. The final follow-up evaluation occurred in December 2012. Patients were eligible for participation if they (1) had at least one of the following Axis-I diagnoses, specified by DSM-IV criteria: Major depressive disorder, social anxiety disorder, panic disorder, generalized anxiety disorder, depressive and/or anxiety disorder not otherwise specified; (2) had a raw score of at least 10 on either the 9-item Patient Health Questionnaire Depression Scale (PHQ-9; *Kroenke, Spitzer & Williams, 2001*) or the 7-item Generalized Anxiety Disorder Scale (GAD-7; *Spitzer et al., 2006*); (3) had no assessed risk of suicidality; (4) had no concurrent psychological treatment that potentially could interfere with the treatment tested; (5) if on psychotropic medication, this treatment had to be stable for three months; (6) did not have other primary disorders that needed different treatments

or that could be affected negatively by the treatment; (7) had no alcohol or drug abuse; (8) were at least 18 years old.

## Randomization and procedure

After initial application, participants were invited to an online screening which consisted of demographic questions and online versions of the outcome measures (see below). These results were later used as a pre-treatment assessment. If initial inclusion criteria were met (having more than 10 on the PHQ-9 or the GAD-7), participants were contacted for a telephone-based diagnostic interview, based on the Mini-International Neuropsychiatric Interview (MINI; *Sheehan et al., 1998*). This procedure is described further below. After confirming additional inclusion criteria, participants were randomized to either treatment or waiting-list (1:1 ratio; block randomization), using an online randomization tool. An independent person, not otherwise involved in the study, handled the randomization. The procedure is illustrated in the CONSORT flowchart in Fig. 1.

## Intervention

The treatment lasted for 10 weeks and consisted of eight self-help modules given with text-based therapist support. A secure online environment was used both for the delivery of self-help material and for communication with the therapists. Therapist support was given asynchronously, i.e., similarly to e-mail. The primary role of the therapists was to give feedback on completed modules and administer gradual access to the treatment. In general, feedback was given on Mondays, but the therapists were available to answer additional questions within 24 h.

The self-help modules were based on the book 'Living Like You Mean It' by Ronald J. Frederick (*Frederick, 2009*) that follows a similar structure as the original affect-phobia treatment manual. Throughout treatment, participants were taught how to practice "emotional mindfulness" as a way of identifying, attending to, and being present with emotional experience. The treatment aimed to teach clients to gradually develop mindful presence as a response to the physical manifestation of emotions which, within the APT model, can be considered as exposure to one's feelings. Throughout the treatment modules, the affect-phobia model as illustrated by the conflict triangle (Fig. 2) was presented to illustrate the function of interventions and to clarify patient case stories. This included techniques to identify and relinquish maladaptive defenses (D), regulate anxiety (A), and approach and experience warded off feelings (F). The final part of the manual contained material on how to make use of experiencing one's core feelings, for example, to express these feelings in interpersonal contexts. In the APT model, expressing feelings to others is seen as essential to shifting both the sense of self and others (*McCullough et al., 2003*). All modules contained homework exercises that needed to be completed before proceeding to the next module. The chapter structure of the manual was: (1) Introduction and problem formulation using the affect-phobia model; (2) Historical understanding and explanation of the problem described; (3) Mindfulness practice to start approaching emotional experience; (4) Defense restructuring; (5) Anxiety regulation techniques; (6) Affect experiencing techniques; (7) Affect expression and self/other restructuring;

(8) A summary of the previous material and advice for continued work. Further details on the treatment can be found in the original treatment manual (*Frederick, 2009*).

## Control group

For ethical reasons, participants on the waiting list also had continuous contact with an assigned therapist during the same 10-week period. Every Monday, therapists were scheduled to initiate contact with the participants, using the same secure online environment as used with the treatment group. Contact involved clinical monitoring of symptoms and questions typically regarding clients' experiences from the previous week. Therapists were instructed to give basic support, but not to use any specific psychological techniques other than empathic listening and asking further questions. As the control group did not work with any treatment modules, the therapists were expected to spend less time with the participants from this group. After the treatment period had ended, participants from the control group were offered an 8-week version of the treatment. The results from that treatment period are, however, outside the scope of this study.

## Outcome measures

The main effect of treatment was assessed using two measures regarding symptoms of depression and anxiety. Depression severity was assessed with the PHQ-9 (*Kroenke, Spitzer & Williams, 2001*), a self-report measure which consists of nine items, each scored 0–3, with a total score ranging from 0 to 27. The PHQ-9 has good psychometric properties, including an internal consistency in the range Cronbach's $\alpha = .86–.89$ and a test-retest reliability of $r = .84$ (*Kroenke et al., 2010*). Several studies have established that the PHQ-9 is sensitive to change during treatment (*Kroenke et al., 2010*). In addition, the PHQ-9 performs similarly regardless of the mode of operation (e.g., as traditional pen and paper, or touch-screen computer; *Fann et al., 2009*). Anxiety severity was measured by the GAD-7 (*Spitzer et al., 2006*), a self-rated 7-item measure, also with items scored 0–3, and with a total score of 21. Internal consistency is excellent (Cronbach's $\alpha = .92$) and with a good test-retest reliability of $r = .83$. Convergent validity of the GAD-7 has been shown to be good, as demonstrated by its correlations with the Beck Anxiety Inventory ($r = .72$) and the anxiety dimension of SCL-90 ($r = .74$) (*Kroenke et al., 2010*). Both measures were administered pre-treatment, weekly during treatment, post-treatment and at the 7-month follow-up.

## Process measures

Two measures were included to assess two processes assumed to be relevant during treatment. Both measures were administered pre-treatment, weekly during treatment, post-treatment and at follow-up. The Emotional Processing Scale (EPS-25; *Baker et al., 2010*) was used to assess emotional processing deficits and the process of emotional change during treatment. In addition, the Swedish 29-item version (*Lilja et al., 2011*) of the Five Facets of Mindfulness Questionnaire (FFMQ; *Baer et al., 2006*) was included to measure the influence of general mindfulness skills. Psychometric properties have been found to be strong for the EPS-25 (Cronbach's $\alpha = 0.92$) (*Baker et al., 2010*) and good for the Swedish

29-item FFMQ (Cronbach's $\alpha = 0.81$) (*Lilja et al., 2011*). The change in total scores on these measures were assumed to reflect an overall change in these processes. A detailed analysis of how these processes were related to treatment outcome will be reported in a separate paper.

## Clinician-administered measures

DSM-IV diagnoses, including a participant's principal diagnosis, were recorded using the MINI Interview (*Sheehan et al., 1998*). This instrument is completely structured, making it suitable for less experienced assessors (*Sheehan et al., 1998*). DSM-IV diagnoses recorded at pre-treatment were followed up at post-treatment and at the 7-month follow-up. The interviewers were blind to treatment condition at post-treatment. Another structured interview was administered at post-treatment and at follow-up, which aimed to give an estimation of global improvement, measured by the 7-point version of the Clinical Global Impression - Improvement (CGI-I) scale (*Guy, 1976*). All interviews were conducted by Master's level final-year clinical psychologist students who were explicitly trained in the diagnostic procedure. A licensed psychologist with a thorough experience from conducting diagnostic interviews provided supervision throughout the assessment period and a psychiatrist was available for additional consultation.

## Therapist training and supervision

The therapists were three Master's level students in their last semester of a 5-year clinical psychologist program. All therapists have had clinical training in affect-focused psychodynamic psychotherapy and had clinical experience from working with this kind of psychotherapy. Prior to the study, all therapists were also trained in providing guided self-help treatments via the Internet. Throughout the trial, clinical supervision was provided by psychologist Ronald J. Frederick, who had authored the original treatment manual. Treatment integrity and adherence to the treatment manual were monitored during supervision.

## Subgroups based on depression and anxiety symptomatology

To investigate differential efficacy between participants who had either depression or anxiety as their main presenting problem, all participants were classified based on their main symptomatology. The classification was based on the assessment of a participant's principal diagnosis that was recorded in the diagnostic interview conducted at baseline. These categories were used to assess whether the treatment was more effective in treating depressive symptoms among participants with principal depression, and analogously regarding anxiety.

## Statistical analyses

Pre-treatment group differences in demographics and on the outcome measures were tested using $\chi^2$-tests (for categorical variables) and independent $t$-tests (for continuous variables). Normality was confirmed by the Shapiro-Wilk test (*Shapiro & Wilk, 1965*) in conjunction with plots of the distribution of data. No significant departure from normality

was detected. Mixed-effects models for repeated-measures data, fitted with maximum likelihood estimation, was used for all continuous outcomes (*Verbeke & Molenberghs, 2000*). Mixed models takes into account all available data from all randomized participants, making it a full intention-to-treat analysis, provides unbiased estimates in the presence of missing data under a fairly unrestrictive missing assumption (i.e., missing at random), and adequately handles nested data structures inherent in repeated-measures data (*Gueorguieva & Krystal, 2004*; *Mallinckrodt, Clark & David, 2001*). All models included random intercepts and slopes, with group, linear time and their interaction included as fixed predictors. Time was considered as a continuous variable and therefore entered as a covariate in the model. For each of the four outcome measures, difference in efficacy between the treatment and the control group were investigated by examining the fixed interaction term of group and linear time. To account for the multiple comparisons, statistical significance was determined using a Bonferroni corrected alpha of 0.0125. Subgroup differences in efficacy were investigated using a fixed three-way interaction term of group, subgroup and time.

Recovery after treatment was defined as having a score less than 10 on both the PHQ-9 and the GAD-7, and not fulfilling criteria for any DSM-IV diagnosis. The same definition was used at follow-up. Between-group differences in recovery at post-treatment were investigated using $\chi^2$-tests. To handle missing data from follow-up diagnostic interviews and estimates of global improvement, post-treatment data were carried forward to the follow-up.

Sample size was determined a priori based on power analyses. These power calculations were based on a linear mixed-effects model (10 time points with an autoregressive error structure with a random intercept and slope), an alpha set at 0.05, power set at 0.80, a predicted effect size of Cohen's $d = 0.50$ and the potential for 10% total attrition rate (at equal rate across time and condition). That analysis suggested that 51.3 participants per group were needed to obtain the desired effect.

Within- and between-group effect sizes (Cohen's $d$) were calculated by dividing the differences in means by the pooled standard deviations (*Borenstein et al., 2009*). Following Cohen's guidelines a between-group effect size in the range of 0.20–0.49 is small, 0.50–0.79 is moderate, and an effect size of 0.80 and above is large (*Cohen, 1988*).

## RESULTS

### Enrollment and baseline characteristics

One hundred individuals with depression and/or anxiety disorders were enrolled in the study. There were no significant pre-treatment mean differences between the treatment group and the control group on any outcome measures (all $t$'s $< 0.97$, all $p$'s $> .33$). Additionally, there were no significant differences between the groups on any demographic data or current/past treatment with medication and/or psychological treatment. A complete description of demographic data of included participants is available in Table 1.

Regarding subgroups of principal depression and anxiety, there was a difference between subgroups in the number of participants in an acute episode of depression,

**Table 1  Demographic description of the participants.**

|  |  | Treatment group | Control group | Total |
|---|---|---|---|---|
| Gender | Male | 8 (16%) | 10 (20%) | 18 (18%) |
|  | Female | 42 (84%) | 40 (80%) | 82 (82%) |
| Age | Mean (SD) | 43.1 (13.9) | 46.6 (12.1) | 44.9 (13.1) |
|  | Min-Max | 19–72 | 23–77 | 19–77 |
| Marital status | Married or co-habiting | 31 (62%) | 36 (72%) | 67 (67%) |
|  | Other | 19 (38%) | 14 (28%) | 33 (33%) |
| Educational level | College or university, at least 3 years | 27 (54%) | 29 (58%) | 56 (56%) |
|  | Other | 23 (46%) | 21 (42%) | 44 (44%) |
| Employment status | Employed or student | 41 (82%) | 33 (66%) | 74 (74%) |
|  | Other | 9 (18%) | 17 (34%) | 26 (26%) |
| Psychological treatment | No experience | 15 (30%) | 16 (32%) | 31 (31%) |
|  | Prior experience | 35 (70%) | 31 (62%) | 66 (66%) |
|  | Ongoing | 0 (0%) | 3 (6%) | 3 (3%) |
| Pharmacological treatment | No experience | 27 (54%) | 22 (44%) | 49 (49%) |
|  | Prior experience | 14 (28%) | 12 (24%) | 26 (26%) |
|  | Ongoing | 9 (18%) | 16 (32%) | 25 (25%) |

$\chi^2(N = 100, df = 1) = 39.4, p < .001$, with 55/57 (96.5%; two participants had depression not otherwise specified) compared to 17/43 (39.5%) for subgroups of depression and anxiety, respectively. Similarly, there were significantly more participants with a principal anxiety disorder that had GAD (67.4%) compared to 35.1% from the depression subgroup, $\chi^2(N = 100, df = 1) = 10.3, p < .001$. There were no differences between subgroups regarding diagnoses of panic disorder and social phobia. Also, there were no differences in any demographics. However, there was a significant difference between subgroups in depression severity as measured by the PHQ-9 at baseline, $t(98) = 3.70, p < .001$. However, no significant baseline difference on the GAD-7 was found $t(98) = 1.23, p = .22$.

## Attrition and adherence

At post-treatment, 100% of the data was collected. At the 7-month follow-up, 47/50 (94%) of the self-report measures and 40/50 (80%) of the data from the follow-up interviews (i.e., diagnostic data and estimates of global improvement) were collected. Adherence to treatment was defined as the number of modules completed. A module was only considered completed if the homework assignment had been sent to the therapist. Out of the 50 participants receiving treatment, 42 (84%) completed all modules. Only 4 participants (8%) completed less than half of the program.

## Outcome and process measures

Means, standard deviations and effect sizes within and between groups for the self-report measures are presented in Tables 2 and 3. Both the treatment group and the control

**Table 2 Means, SDs and effect sizes (Cohen's d) for measures of depression and anxiety.**

| Measure and group | Mean (SD) | | | Effect size Cohen's d (95% CI) | | | Linear mixed models | |
|---|---|---|---|---|---|---|---|---|
| | Pre-treatment | Post-treatment | 7-month follow-up | Between-group, post-treatment | Within-group pre-post-treatment | Within-group pre-7-month follow-up | Effect | p |
| **PHQ-9** | | | | | | | | |
| Treatment group (n = 50) | 13.90 (3.6) | 6.32 (4.2) | 5.55 (3.5) | 0.77 (0.37–1.18) | 1.93 (1.31–2.55) | 2.43 (1.72–3.14) | Group | .89 |
| Depression subgroup (n = 28) | 15.32 (3.3) | 5.89 (2.8) | 5.96 (3.5) | 0.95 (0.40–1.50) | 3.10 (1.87–4.32) | 2.82 (1.78–3.87) | Time | <.001 |
| Anxiety subgroup (n = 22) | 12.09 (3.3) | 6.86 (5.5) | 5.00 (3.4) | 0.55 (−0.06–1.16) | 1.12 (0.49–1.75) | 2.17 (1.11–3.24) | G × T | <.001 |
| Control group (n = 50) | 13.96 (4.7) | 10.26 (5.9) | | | 0.69 (0.40–0.97) | | | |
| Depression subgroup (n = 29) | 15.07 (4.4) | 10.59 (6.4) | | | 0.79 (0.37–1.22) | | | |
| Anxiety subgroup (n = 21) | 12.43 (4.7) | 9.81 (5.2) | | | 0.53 (0.20–0.85) | | | |
| **GAD-7** | | | | | | | | |
| Treatment group (n = 50) | 11.46 (4.0) | 6.12 (4.5) | 5.34 (4.1) | 0.48 (0.08–0.87) | 1.25 (0.79–1.71) | 1.51 (0.97–2.06) | Group | .85 |
| Depression subgroup (n = 28) | 10.86 (4.1) | 5.46 (3.9) | 5.19 (4.1) | 0.56 (0.03–1.09) | 1.35 (0.75–1.95) | 1.43 (0.73–2.10) | Time | <.001 |
| Anxiety subgroup (n = 22) | 12.23 (3.8) | 6.95 (5.3) | 5.55 (4.2) | 0.39 (−0.21–0.99) | 1.15 (0.44–1.86) | 1.62 (0.72–2.52) | G × T | <.01 |
| Control group (n = 50) | 12.26 (4.2) | 8.40 (5.0) | | | 0.82 (0.51–1.13) | | | |
| Depression subgroup (n = 29) | 11.97 (5.0) | 8.03 (5.3) | | | 0.76 (0.39–1.13) | | | |
| Anxiety subgroup (n = 21) | 12.67 (2.8) | 8.90 (4.7) | | | 0.93 (0.36–1.50) | | | |

**Notes.**

The confidence intervals were calculated using the standard error and an alpha level of 0.05. A confidence interval that do not overlap zero indicates a significance of $p < .05$. $p$ values are given for linear mixed models using data from pre-treatment and post-treatment and all individuals ($N = 100$). The $p$ value associated with the main effect of group denotes significance of average difference between groups at the pre-treatment assessment. The $p$ value associated with the effect of time denotes the significance of average change over all assessment periods across treatment. The $p$ value associated with the effect of G × T (group × time) denotes significance of difference between the treatment group and the control group in change over all assessment periods. Effect sizes are calculated as standardized mean differences. PHQ-9: 9-item Patient Health Questionnaire Depression Scale; GAD-7: 7-item Generalized Anxiety Disorder Scale.

**Table 3 Means, SDs and effect sizes (Cohen's d) for measures of emotional processing and mindfulness skills.**

| Measure and group | Mean (SD) | | | Effect size Cohen's d (95% CI) | | | Linear mixed models | |
|---|---|---|---|---|---|---|---|---|
| | Pre-treatment | Post-treatment | 7-month follow-up | Between-group, post-treatment | Within-group pre-post-treatment | Within-group pre-7-month follow-up | Effect | p |
| **EPS-25** | | | | | | | | |
| Treatment group (n = 50) | 5.00 (1.03) | 2.86 (1.48) | 2.84 (1.65) | 0.82 (0.41–1.23) | 1.67 (1.13–2.21) | 1.51 (1.00–2.01) | Group | .50 |
| | | | | | | | Time | <.001 |
| Control group (n = 50) | 4.93 (1.01) | 4.17 (1.73) | | | 0.50 (0.22–0.77) | | G × T | <.001 |
| **FFMQ** | | | | | | | | |
| Treatment group (n = 50) | 76.70 (10.9) | 88.00 (12.0) | 88.98 (13.3) | 0.65 (0.25–1.05) | 0.98 (0.65–1.31) | 0.99 (0.59–1.39) | Group | .15 |
| | | | | | | | Time | <.001 |
| Control group (n = 50) | 77.18 (14.1) | 78.44 (17.1) | | | 0.08 (−0.11–0.27) | | G × T | <.001 |

**Notes.**

The confidence intervals were calculated using the standard error and an alpha level of 0.05. A confidence interval that do not overlap zero indicates a significance of $p < .05$. $p$ values are given for linear mixed models using data from pre-treatment and post-treatment and all individuals ($N = 100$). The $p$ value associated with the main effect of group denotes significance of average difference between groups at the pre-treatment assessment. The $p$ value associated with the effect of time denotes the significance of average change over all assessment periods across treatment. The $p$ value associated with the effect of G × T (group × time) denotes significance of difference between the treatment group and the control group in change over all assessment periods. Effect sizes are calculated as standardized mean differences. EPS-25: Emotional Processing Scale; FFMQ: Five Facets of Mindfulness Questionnaire.

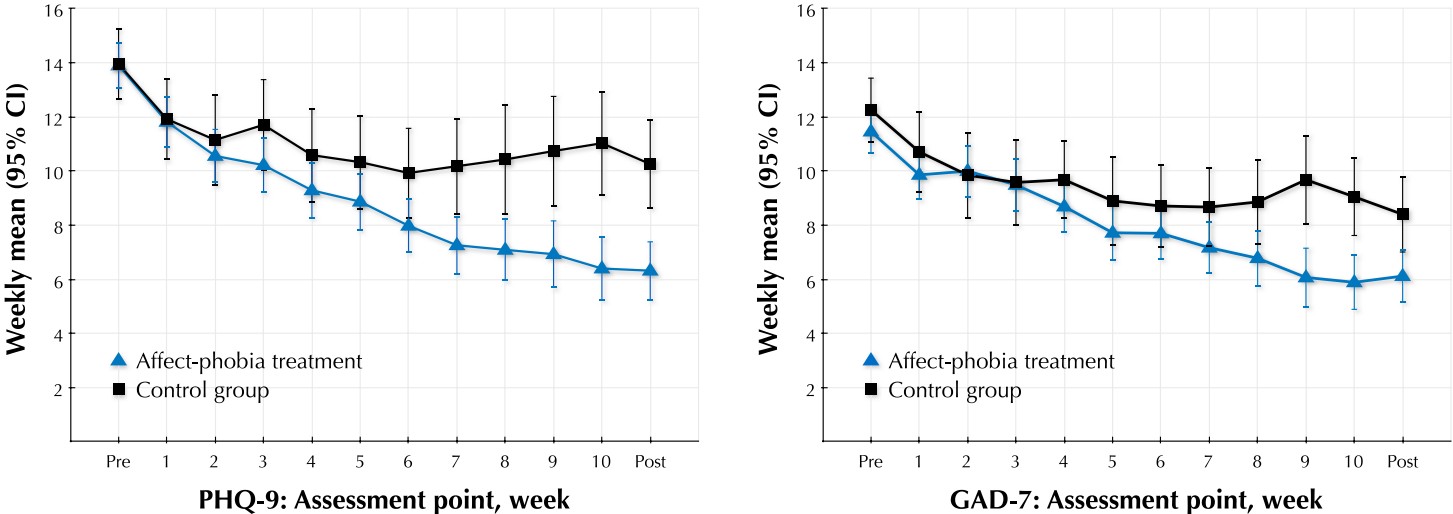

**Figure 3  Weekly PHQ-9 and GAD-7 scores.** Weekly scores on the PHQ-9 and the GAD-7 for both groups. Vertical bars denote 95% confidence intervals (CI). PHQ-9: 9-item Patient Health Questionnaire Depression Scale; GAD-7: 7-item Generalized Anxiety Disorder Scale.

group had substantial within-group effects after the 10-week period. Mixed models analyses revealed significant interaction effects of treatment group and time on the PHQ-9, $F(1, 102.1) = 19.94, p < .001$, and the GAD-7, $F(1, 105.1) = 7.86, p < .01$. Both interaction effects were significant at the Bonferroni-corrected alpha level of $p < .0125$. Estimates of fixed effects for the PHQ-9 and the GAD-7 were as follows. Intercept: 12.37 and 10.71 ($p$'s $< .001$); Group: 0.11 and 0.14 ($p$'s $> .85$); Time: $-0.64$ and $-0.49$ ($p$'s $< .001$); Group $\times$ Time: 0.40 and 0.23 ($p$'s $< .01$). Random effects for the PHQ-9 and the GAD-7 were estimated as follows. Intercept: 14.33 and 12.78 ($p$'s $< .001$); Time: 0.15 and 0.12 ($p$'s $< .001$). Between-group effect sizes at post-treatment was large ($d = 0.77$) for depression and moderate ($d = 0.48$) for anxiety, favoring treatment over control. The continuous within-group changes on the PHQ-9 and the GAD-7 are illustrated in Fig. 3. At the 7-month follow-up, the treatment effect was stable. Paired $t$-tests conducted post hoc showed that there were significant post-treatment versus follow-up decrease on the GAD-7, $t(46) = 2.03, p < .05$, and a trend towards a significant decrease on the PHQ-9, $t(46) = 1.42, p = .16$. For the EPS-25 and the FFMQ, there were also significant interaction effects of treatment group and time ($F(1, 104.5) = 26.5$ and $F(1, 101.2) = 29.9$, respectively; Both $p$'s $< .001$). Both these interaction effects were significant at the Bonferroni-corrected alpha level of $p < .0125$. Fixed effects for the EPS-25 and the FFMQ were estimated as follows. Intercept: 4.70 and 72.76 ($p$'s $< .001$); Group: $-0.16$ and 3.67 ($p$'s $> .15$); Time: $-0.18$ and 1.39 ($p$'s $< .001$); Group $\times$ Time: 0.13 and $-1.17$ ($p$'s $< .001$). Estimates of random effects for the EPS-25 and the FFMQ were as follows. Intercept: 1.25 and 156.37 ($p$'s $< .001$); Time: 0.013 and 0.98 ($p$'s $< .001$). The between-group effect at post-treatment was large for the EPS-25 ($d = 0.82$) and moderate to large ($d = 0.65$) for the FFMQ.

**Table 4 Frequency data of DSM-IV diagnoses.**

| Diagnosis | Treatment group | | | Control group | |
|---|---|---|---|---|---|
| | Pre-treatment | Post-treatment | 7-month follow-up | Pre-treatment | Post-treatment |
| DEP | 35 (70%) | 5 (10%) | 6 (12%) | 37 (74%) | 16 (32%) |
| GAD | 23 (46%) | 13 (26%) | 10 (20%) | 26 (52%) | 18 (36%) |
| SP | 19 (38%) | 10 (20%) | 9 (18%) | 17 (34%) | 13 (26%) |
| PD | 11 (22%) | 5 (10%) | 6 (12%) | 9 (18%) | 5 (10%) |
| **Number of diagnoses** | | | | | |
| 0 | 4 (8%) | 28 (56%) | 31 (62%) | 0 (0%) | 18 (36%) |
| 1 | 16 (32%) | 13 (26%) | 8 (16%) | 23 (46%) | 18 (36%) |
| 2 | 19 (38%) | 7 (14%) | 10 (20%) | 16 (32%) | 9 (18%) |
| 3 | 10 (20%) | 2 (4%) | 1 (2%) | 10 (20%) | 4 (8%) |
| 4 | 1 (2%) | 0 (0%) | 0 (0%) | 1 (2%) | 1 (2%) |
| Total number of diagnoses | 88 | 33 | 31 | 89 | 49 |

**Notes.**
The four participants with zero diagnoses listed at pre-treatment fulfilled DSM-IV criteria for depression and anxiety, not otherwise specified. DEP, GAD, SP, PD: Diagnoses of major depression, generalized anxiety disorder, social phobia and panic disorder.

## Diagnoses

The number of diagnoses among participants at pre-treatment, post-treatment and at the 7-month follow-up are illustrated in Table 4. At post-treatment, there were significantly fewer participants with a diagnosis of major depression in the treatment group (10%) than in the control group (32%). The difference was significant ($\chi^2(N = 100, df = 1) = 7.3$, $p < .01$). Reductions in the number of diagnoses of GAD, SP or PD were not significantly different between groups at post-treatment.

## Recovery after treatment and clinical global improvement

Categorical rates of recovery after treatment (i.e., a participant who did not fulfill criteria for any DSM-IV diagnosis and reached a score less than 10 on both the PHQ-9 and the GAD-7) were significantly different at post-treatment between the treatment group ($n = 26$; 52.0%) and the control group ($n = 12$; 24.0%), $\chi^2(N = 100, df = 1) = 8.3$, $p < .01$. At follow-up there were 25 participants (50.0%) from the treatment group who met the criteria for recovery.

Post-treatment interviews resulted in estimates of clinical global improvement according to the CGI-I (*Guy, 1976*). In the treatment group, 28 participants (56.0%) were much or very much improved while this was only true for 11 (22.0%) in the control group. This difference was significant, $\chi^2(N = 100, df = 1) = 12.1$, $p < .001$. At follow-up, this figure was 52% ($n = 26$) in the treatment group.

## Subgroups of principal depression and anxiety

There were no significant interaction effect of group, subgroup and time on neither the PHQ-9 nor the GAD-7. This was despite the fact that the treatment had a very large within-group group effect ($d = 3.10$) on the PHQ-9 in the depression subgroup, compared

to for those in the anxiety subgroup ($d = 1.12$). Thus, there were no indications that the treatment was more effective in reducing symptoms of depression among participants with a principal diagnosis of depression, or analogously for anxiety symptoms.

### Therapist time

In the treatment group, the average therapist time per client and week was 9.5 min ($SD = 4.0$). While there was a significant difference in average therapist time per week between therapists ($F(2, 47) = 7.73, p < .001$), there were no correlations between therapist time and change scores on any of the outcome measures (all $r$'s $< .19$, all $p$'s $> .18$). The average therapist time per client and week was 2.3 min ($SD = 0.86$) in the control group. As expected, therapist time was significantly less in the control group than in the treatment group, $t(98) = 12.4, p < .001$.

### Participants' evaluation of the treatment

Most participants were satisfied (46%) or very satisfied (36%) with the overall treatment they had received. Nine (18%) were indifferent or mildly dissatisfied, and no one was clearly dissatisfied. An absolute majority (82%) thought that the amount of text was appropriate. A similar amount of participants considered the text interesting and relevant, all the time (46%) or most of the time (40%). Most participants considered the treatment to be very demanding (28%), demanding (42%) or somewhat demanding (26%). Importantly though, a majority considered the treatment very much worth the effort (52%) or worth the effort (38%).

### DISCUSSION

This randomized controlled trial aimed to evaluate the effects of affect-phobia therapy in the format of guided self-help through the Internet in a sample of participants with depression and anxiety disorders. The results indicated that the treatment was effective in reducing symptoms of depression and anxiety, and also in facilitating emotional processing and mindfulness skills. Subgroup analyses gave no indications of differential efficacy between participants with a principal diagnosis of depression and those with principal anxiety. Treatment gains were maintained in the 7-month follow-up.

The treatment manual used in this study aimed to implement a psychodynamic treatment based on the affect-phobia model in self-help format. This approach calls for a discussion on similarities and differences to the original APT manual. An assumption of this implementation was that the core principles of affect-phobia treatment manual could be retained. This included the general model of psychopathology (i.e., as illustrated by the triangle of conflict in Fig. 2) and the overall structure of the therapy. While the treatment emphasized how affect-phobic patterns in a person's current life (C) began with past persons (P), as illustrated in Malan's triangle of person, the treatment did not address how these patterns could potentially be enacted with the therapist (T). Importantly, these patterns were not regarded as non-existing, but rather the treatment material did not address them nor was it part of the role of the therapists to address these patterns. While the therapist role might overlap between guided self-help and face-to-face therapy in

several aspects (*Paxling et al., 2013*), there is a difference in the present study in how the treatment material taught "emotional mindfulness" as a way of conducting exposure to one's feelings without the therapist being present. Some authors have suggested that exposure with response prevention may result in better effects of treatment when patients conduct the exposure by themselves, in their natural environment (*Röper & Rachman, 1976*; *Salkovskis, 1985*). If this is also the case in affect-phobia therapy and how that would affect outcome is a question for further research, but it is possible that self-exposure to feelings is at least as effective as exposure with a therapist present. Importantly, as exposure with response prevention is assumed to be an active mechanism in the treatment tested in this study, this would imply similarities in working mechanisms in affect-focused psychodynamic treatment and cognitive behavioral therapy based on principles of exposure. Similarly, several contemporary CBT treatments have components of emotion regulation techniques (e.g., *Berking et al., 2013*; *Bryant et al., 2013*), that seem overlapping to affect-phobia treatment. Future research should investigate further similarities and differences between affect-focused psychodynamic therapy and various CBT models. Summing up, despite the aforementioned differences to the original APT manual, we believe that the manual used in the current study is indeed a valid implementation of a psychodynamic therapy based on the affect-phobia model.

In affect-phobia therapy, the model of psychopathology is the same across disorders, i.e., the triangle of conflict is assumed to explain both etiology and maintenance of for example depression and anxiety disorders (*McCullough et al., 2003*). This aim is similar to transdiagnostic and unified protocols where the treatment material has been arranged to fit a broader range of patients (*Barlow, Allen & Choate, 2004*; *Craske, 2012*). Hence, affect-phobia therapy could be described as a transdiagnostic treatment. While there are several studies on the efficacy of cognitive behavioral transdiagnostic treatments for anxiety disorders (*Farchione et al., 2012*; *McEvoy, Nathan & Norton, 2009*), few exist that explicitly target both depression and anxiety. However, one uncontrolled trial testing the effectiveness of a group-based intervention (*McEvoy & Nathan, 2007*) resulted in promising outcomes and showed comparable efficacy to several disorder-specific treatments. More recently, *Titov et al. (2011)* provided evidence of the efficacy of an Internet-delivered transdiagnostic program that targeted both anxiety and depression, when compared to a waiting-list. Both these treatments yielded within-group effect sizes of Cohen's *d* around 1.0 for measures of depression and anxiety. Hence, the affect-phobia treatment tested in this study, seems to stand well when compared to other transdiagnostic treatments tested.

There are methodological limitations that need to be considered. First, as we recruited participants from the community and not from, for example, a treatment clinic, the external validity of the findings are challenging to interpret. While there are studies on ICBT that suggests generalizability to clinical settings (e.g., *Bergström et al., 2010*; *Hedman et al., 2013*), this has yet to be proven for Internet-delivered psychodynamic therapy. Moreover, more than half of the participants in the present study had three years or more of university education. While this factor might have biased the results, the average severity

of depression and anxiety symptoms was moderate to moderately severe (*Kroenke et al., 2010*), and more than half of the participants had comorbid disorders, suggesting clinical representativity (*Kessler, Merikangas & Wang, 2007*). A second methodological limitation concerns the substantial within-group effects in the control group that make the results harder to interpret. These effects are probably due to the weekly clinical monitoring and supportive contact with the therapists in addition to the extensive test procedures such as telephone interviews before and after the treatment period. While these aspects might have biased the results, it also highlights the need for research regarding specific factors in guided self-help treatments. A third limitation concerns how missing data were handled at the 7-month follow-up for the categorical measure of recovery. For that measure, missing data at follow-up were replaced by that from post-treatment. When using that approach in longitudinal analysis, there is a known risk for estimation bias (*Gueorguieva & Krystal, 2004*). While we did not compare the recovery rates over time (i.e., did not include this measure in any longitudinal analysis) we still acknowledge that some method of multiple imputation could have been used to account for the missing data (*Graham, 2009*). Another limitation concerns the subgroup analyses. As the total sample size was appropriate for the comparison between treatment and control, it seemed not to have been adequate for the analyses regarding subgroups of principal depression and anxiety. This implies that a larger sample size would be needed in future research if investigating the effect of a transdiagnostic treatment on specific diagnoses in a sample with multiple disorders. A related limitation concerns the definition of the subgroups in this study. The group with principal anxiety disorders is a heterogeneous group, e.g., in the sense that it includes both fear disorders (e.g., panic disorder) and worry disorders (GAD). This may have confounded the subgroup analyses. A final limitation that needs to be addressed concerns the therapists in the study, all of whom were psychologists in training, albeit during the last semester of training in a five year program and under regular supervision. It is possible that more experienced therapists would have enabled even larger treatment effects. A related concern is that psychologists in training conducted all diagnostic interviews. While the psychologists were explicitly trained in the diagnostic procedures and received supervision, there is a possibility that their level of experience may have affected how the diagnostic categories were defined. Importantly though, the MINI interview has been designed to be administered by non-experts.

## CONCLUSIONS

This study provides preliminary support for the efficacy of Internet-delivered psychodynamic treatment based on the affect-phobia model in the treatment of depression and anxiety disorders. This study provides further evidence that psychodynamic treatment approaches may be transferred to the guided self-help format and delivered via the Internet. Hence, this study adds to the empirical base of Internet-delivered psychological treatments and to that of psychodynamic psychotherapy in general. Finally, as we have no reason to believe that the treatment would perform less effectively in a face-to-face setting, the findings from this study call for further research on affect-focused psychotherapies.

## ACKNOWLEDGEMENTS

We would like to thank Frida Forsman, Linda Karlgren and Anton Sandell for conducting diagnostic interviews at post-treatment and at follow-up, and in addition acted as therapists when providing treatment to the control group. Additional thanks to Maximilian Rubinsztein for conducting pre-treatment interviews and to Peter Lilliengren for valuable comments on the manuscript. We would also like to thank Per Carlbring for help during the recruitment phase and Alexander Alasjö for technical support. We also thank Linköping University for funding and the Internet psychiatry unit in Stockholm, Sweden for the use of the treatment platform. Finally, we would also like to acknowledge the participants for their involvement and helpful comments.

### Funding

The study was supported in part by the Swedish Council for Working and Life Research and by Linköping University (Professor contract extended to Gerhard Andersson). The funders had no role in study design, data collection and analysis, decision to publish, or preparation of the manuscript.

### Grant Disclosures

The following grant information was disclosed by the authors:
Swedish Council for Working and Life Research.
Linköping University.

### Competing Interests

Professor Andersson is an Academic Editor for PeerJ. Dr. Frederick is the author of the book used as a basis for the treatment manual. Otherwise the authors have no competing interests.

### Author Contributions

- Robert Johansson, Martin Björklund, Christoffer Hornborg, Stina Karlsson, Hugo Hesser and Gerhard Andersson conceived and designed the experiments, performed the experiments, analyzed the data, wrote the paper.
- Brjánn Ljótsson conceived and designed the experiments, performed the experiments, wrote the paper.
- Andréas Rousseau conceived and designed the experiments.
- Ronald J. Frederick conceived and designed the experiments, wrote the paper.

### Clinical Trial Ethics

The following information was supplied relating to ethical approvals (i.e., approving body and any reference numbers):

This study received approval from the Regional Ethics Board of Linköping, Sweden (Approval number: 2011/400-31).

## Clinical Trial Registration

The following information was supplied regarding Clinical Trial registration:

Clinicaltrials.gov registration ID is NCT01532219.

## Supplemental Information

Supplemental information for this article can be found online at http://dx.doi.org/
10.7717/peerj.102.

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
