# Peer review of "Affect-focused psychodynamic psychotherapy for depression and anxiety through the Internet: a randomized controlled trial"

_PeerJ, doi:10.7717/peerj.102_

## Round 0.1 · original submission · Minor Revisions

Thank you for your submission to Peer J, as you will observe, the reviewers comments are largely positive and I look forward to receiving your revision

Reviewer 1 ·

Basic reporting

Johansson et al report conducted a small clinical trial to investigate the efficacy of an Internet-based, psychodynamic, guided self-help treatment based on APT for depression and anxiety disorders. The paper is well written and the statistical are ok by they need to be improved.

Experimental design

The desing is a simple parallel design placebo controlled randomized trial

Validity of the findings

The findings seem to be ok, taking into account the limitations of the nalyses and the limitations the authors acknowledged themself in the discussion. I have made several comments below which are aimed at improving the clarity of the paper and mainly the reporting of the statistical methodology and results.

Additional comments

Abstract
Background: Please explain briefly what is "Axis-I disorders"
Methods: Please explain what is DSM-IV
Methods: Second line I would write "active group versus control" , instead of what iscurrently written
Introduction: Page 2, First paragraph, 4th line the figures 20.8% and 28.8% please explain if these are figures for a specific country or if they are world wide figures.

Introduction
Page 3: First paragraph, please explain the rational of using the word "client" instead of patient.
Page 4: The last paragraph, why the analysis/comparison is based on week 10 and baseline only? and not on the trajectory of change over the entire treatment period.
Page6: This is a major issue in this paper, I mean multiplicity, the authors have investigated a number of outcomes, therefore, there is a clear multiplicity problem which the authors should discuss and address.
Page 8: Outcome measure, line 10, it is not clear why self-rated 7-item measure can add-up to 18 and not 21.
Page11: Statistical analyses. The authors used t-test for continuous variables which assumes normality, what about skewed variables? other statistical test are more suitable for those variables. Also please specify that X2 test was used for categorical variables and t-test for continuous ones.
Page 11: Statistical analyses, second paragraph. The authors used post-treatment carrid forward for imputation, what is the rational of choosing this approach? please discuss the potential limitations of this imputation approach and consider alternative approach as a sensitivity analysis.
Page12: The authors did not discuss the distributions of their outcomes, I suggest they provide histograms for the outcomes used in the analyses.
Page 12: Results line 2, please specify that the comparison concerns the means , because the outcome may be diffirent betwen the two groups in terms of other aspects of its distribution, variance for example.
Page13: Outcome and process measures: the authors should clarify how time was analyzed, discrete or continuous (which seems to be the case) and explain why using one aproach over the other.
Page 15: When using mixed model, I would expect reporting random effects, and Intra class correlation (ICC) these seem to be lacking from this paper.
Table2: Please provide p values together with confidence intervals.

Reviewer 2 ·

Basic reporting

See below

Experimental design

See below

Validity of the findings

See below

Additional comments

The authors have carried out a randomized controlled trial comparing affect-focused psychodynamic psychotherapy for depression and anxiety to a control condition through the internet. Psychodynamic internet-delivered therapy was based on McCullough’s model of affect phobia. This model was transferred to self-help modules that were based on a book by Frederik that is compatible with the McCullough approach. Patients in the control condition were also offered online contact with the psychotherapist but did not receive the self-help modules. Treatments were carried out by advanced students with some clinical experience and training in internet-based therapy and in the McCullough treatment. The study included both patients with depression and anxiety disorders. Outcome was assessed by the 9e-item Patient Help Questionnaire Depressions Scale (PHQ-9) and the 7-item Generalized Anxiety Disorder Scale (GAD-7). Furthermore, recovery was assessed and process measures were included.

The psychodynamic internet-delivered treatment proved to be significantly superior to the control condition with regard to both the PHQ-9 and the GAD-7I. This was also true for rates of recovery.

In all, the study is well designed and conducted. Minor revisions are recommended:

On page 4 the authors present the study hypothesis. With regard to the hypothesis the authors are recommended to directly refer to the comparison of the treatments, not only to the within-group effects.


Concerning recovery as assessed by diagnosis, the psychodynamic treatment was significantly superior to the control condition with regard to the diagnosis of major depression. However, the psychodynamic treatment was not superior to the control condition with regard to reducing the number of diagnoses of anxiety disorders. The authors present the data on recovery after treatment defined by the criterion of not for filling any DSM-IV diagnosis and a score of less than 10 on both the PHQ-9 and the GAD-7. The difference in favour of the psychodynamic treatment was significant. However, the authors did not report results for depressive and anxiety disorders separately. It can be expected from the data in Table 4 that again differences with regard to recovery from anxiety disorders are not significant. The authors are recommended to add these data in the text. With regard to continuous measures of depression and anxiety the authors did not find differential efficacy for depressive and anxiety disorders. However, as mentioned above the psychodynamic treatment was not significantly superior to the control condition with regard to rates of recovery. The authors should discuss this result more in detail.

Furthermore they found significant differences between the treatments with regard to the time per client. The authors are recommended to discuss this result more in detail as well. As I have understood from the design, the control condition was not planned to receive less contacts with the therapist.

If the authors take these issues discuss above adequately into account the article can be recommended for publication.

·

Basic reporting

Overall, this is a very well written article.

Relevant prior literature is appropriately described and referenced and the case for this study is well made. The structure of the article is clear, clean, and concise.

The authors should review the sentence beginning on line 10, p15, which would benefit from editing.

Experimental design

This paper represents original primary research, the research question is well defined, and the study reflects a high technical standard.

The sample size calculations were appropriate to address the overall comparison between treatment and control, but were not adequate for the subgroup analyses. This limitation should be acknowledged in the Discussion. Moreover, it is conceptually problematic to include fear disorders (panic/social anxiety) with worry disorders (GAD), and this may have further confounded subgroup analyses. Again, this should be acknowledged in the Discussion.

Validity of the findings

The data appear robust and statistically sound. The exclusion of participants with low or high scores helped restrict standard deviation, which increased within group effect sizes, but the authors appropriately focused on between-group effect sizes as indicators of difference due to treatment.

The control group appeared to receive a non impotent intervention, which increases confidence that the treatment intervention was efficacious.

The conclusion that the manual used in the study represented a 'solid implementation' of a psychodynamic therapy based on the affect-phobia model (pg 17, line 17) is difficult to evaluate. I suggest the authors be more circumspect about this position.

Additional comments

The components of the treatment appear common to evidence-based principles of recovery from anxiety/depression. I encourage the authors to consider this issue in the Discussion.

---

## Round 0.2 · accepted · Accept

Dear Dr Johansson, Many thanks for your rapid revision. All the points raised during the review have been addressed.